# Bioengineering Approaches to Improve In Vitro Performance of Prepubertal Lamb Oocytes

**DOI:** 10.3390/cells10061458

**Published:** 2021-06-10

**Authors:** Antonella Mastrorocco, Ludovica Cacopardo, Daniela Lamanna, Letizia Temerario, Giacomina Brunetti, Augusto Carluccio, Domenico Robbe, Maria Elena Dell’Aquila

**Affiliations:** 1Faculty of Veterinary Medicine, University of Teramo, Loc. Piano d’Accio, 64100 Teramo, Italy; acarluccio@unite.it (A.C.); drobbe@unite.it (D.R.); 2Research Centre E. Piaggio, University of Pisa, Largo Lucio Lazzarino 1, 56122 Pisa, Italy; ludovica.cacopardo@ing.unipi.it; 3Department of Biosciences, Biotechnologies & Biopharmaceutics, University of Bari Aldo Moro, Via Edoardo Orabona, 70125 Bari, Italy; daniela.lamanna95@gmail.com (D.L.); letiziatem@gmail.com (L.T.); giacomina.brunetti@uniba.it (G.B.); mariaelena.dellaquila@uniba.it (M.E.D.)

**Keywords:** prepubertal lamb oocyte, JIVET, 3D in vitro maturation, millifluidic bioreactor, bioprinting, granulosa cells, collagen I, oocyte bioenergetics, computational fluidodynamic model, convection–diffusion–reaction model

## Abstract

Juvenile in vitro embryo technology (JIVET) provides exciting opportunities in animal reproduction by reducing the generation intervals. Prepubertal oocytes are also relevant models for studies on oncofertility. However, current JIVET efficiency is still unpredictable, and further improvements are needed in order for it to be used on a large-scale level. This study applied bioengineering approaches to recreate: (1) the three-dimensional (3D) structure of the cumulus–oocyte complex (COC), by constructing—via bioprinting technologies—alginate-based microbeads (COC-microbeads) for 3D in vitro maturation (3D-IVM); (2) dynamic IVM conditions, by culturing the COC in a millifluidic bioreactor; and (3) an artificial follicular wall with basal membrane, by adding granulosa cells (GCs) and type I collagen (CI) during bioprinting. The results show that oocyte nuclear and cytoplasmic maturation, as well as blastocyst quality, were improved after 3D-IVM compared to 2D controls. The dynamic 3D-IVM did not enhance oocyte maturation, but it improved oocyte bioenergetics compared with static 3D-IVM. The computational model showed higher oxygen levels in the bioreactor with respect to the static well. Microbead enrichment with GCs and CI improved oocyte maturation and bioenergetics. In conclusion, this study demonstrated that bioengineering approaches that mimic the physiological follicle structure could be valuable tools to improve IVM and JIVET.

## 1. Introduction

Oocytes from prepubertal lambs represent a unique model to study oocyte modifications occurring in the period between birth and puberty. These oocytes are used for JIVET (juvenile in vitro embryo technologies), a promising set of assisted reproductive technologies (ARTs) providing exciting opportunities in different application fields of animal reproduction. Indeed, JIVET increases genetic gain and reproductive efficiency while shortening the generation interval in breeding programs [1,2,3,4,5], in the logic of sustainable animal breeding and preservation of bioresources, such as the germplasm of native or endangered animal breeds and species. Moreover, prepubertal lamb oocytes are of relevant scientific interest as models of oocytes recovered at pediatric and adolescent age in humans [6]. Finally, another advantage of oocyte collection from prepubertal ovaries is the availability of large numbers of oocytes from non-atretic follicles [2].

It has been demonstrated that oocytes from prepubertal lambs progress through meiosis to metaphase II (MII) at rates similar to oocytes from adult animals, and can be fertilized, developed, and give rise to viable blastocysts [7,8,9], which can then be transferred to adult recipients, ultimately producing live offspring [10,11,12]. Nevertheless, the greatest limitation of prepubertal lamb oocytes’ use in JIVET is that their developmental competence is lower in comparison to oocytes derived from their adult counterpart [13]—a phenomenon also observed in other species [13,14,15,16,17]. This lower embryo developmental competence of oocytes from juvenile ewes has to date been mostly related to incomplete or perturbed in vitro cytoplasmic, molecular, and nuclear maturation. In detail, it has been shown that oocytes from prepubertal animals exhibit ultrastructural, molecular, and functional differences compared to adult ones [18,19,20,21,22,23,24,25,26,27,28,29,30,31,32].

Despite considerable research efforts in recent years, the current efficiency of the JIVET approach is still unpredictable and variable, and further improvements are needed before large-scale commercial use. To date, several attempts have been made to enhance the developmental competence of prepubertal lamb oocytes, via several strategies, such as media formulations [33,34] or donor selection by the use of predictive endocrine biomarkers [35]. To the best of our knowledge, no studies have been performed using bioengineering approaches to date.

We recently published a three-dimensional in vitro maturation (3D-IVM) protocol in which the cumulus–oocyte complex (COC) was encapsulated inside the core of an alginate microbead by a one-step automated bioprinting-based method and further cultured under 3D conditions. Thanks to this method, nuclear and cytoplasmic maturation of adult sheep oocytes were significantly increased [36]. Using this approach, integrated with additional technological improvements, the aim of the present study was to improve the in vitro performance of prepubertal lamb oocytes. Physiological follicular conditions were reproduced in vitro with the purposes of: (1) recreating the three-dimensional (3D) structure of the COC; (2) establishing millifluidic culture conditions; and (3) enhancing typical interactions between the COC and granulosa cells (GCs) by using biomaterials able to mimic the composition of the follicular extracellular matrix (ECM).

## 2. Materials and Methods

### 2.1. Chemicals

All chemicals for in vitro cultures and analyses were purchased from Sigma-Aldrich (Milan, Italy), unless otherwise indicated.

### 2.2. Collection of Ovaries and COC Retrieval

Ovaries from prepubertal lambs (under 6 months of age) were recovered from local slaughterhouses. Ovaries were transported to the laboratory at room temperature within 4 h after collection. For COC retrieval, ovaries underwent the slicing procedure [37]. Follicular contents were released in sterile Petri dishes containing phosphate-buffered saline (PBS) supplemented with 1 mg/mL heparin and observed under a Nikon SMZ18 stereomicroscope equipped with a transparent heating stage set up at 37 °C (Okolab, Napoli, Italy). Only COCs with at least three intact cumulus cell (CC) layers and homogenous cytoplasms were selected for culturing [35].

### 2.3. In Vitro Maturation (IVM) Medium

IVM medium was prepared based on TCM-199 medium with Earle’s salts, and was buffered with 5.87 mmol/L HEPES and 33.09 mmol/L sodium bicarbonate, and supplemented with 0.1 g/L l-glutamine, 2.27 mmol/L sodium pyruvate, calcium lactate pentahydrate (1.62 mmol/L Ca^2+^, 3.9 mmol/L Lactate), 50 μg/mL gentamicin, 20% (*v*/*v*) fetal calf serum (FCS), 10 μg/mL of porcine follicle-stimulating hormone (FSH) and luteinizing hormone (LH; Pluset^®^, Calier, Barcelona Spain) [38], and 1 μg/mL 17β estradiol [37]. IVM medium was pre-equilibrated for 1 h under 5% CO_2_ in air at 38.5 °C.

### 2.4. Alginate COC-Microbead Fabrication

Alginate microbeads including COCs (COC-microbeads) were fabricated using the spherical hydrogel generator (SpHyGa) [36,39]. Briefly, the SpHyGa was loaded with a syringe filled with a 1% sodium alginate solution in IVM medium and fitted with a needle with a 508 μm internal diameter, adapted to ovine COC size [36]. COCs were added to the alginate solution by using a Gilson-type pipette with sterile tips. The stepper motor driving the syringe plunger was set to a constant flow rate of 10 μL s^−1^ through the graphical user interface. At this flow rate, roughly one drop per COC was generated on the needle tip. The droplet landed in a beaker filled with a 0.1 mol/L CaCl_2_ crosslinking solution, whereupon the droplet formed alginate COC-microbeads—a hydrogel bead in which the COC was enclosed. COC-microbeads were collected and kept immersed in the CaCl_2_ solution to allow homogeneous crosslink formation. The microbeads were then observed under an optical stereomicroscope and only those containing one COC were selected and used for IVM.

### 2.5. Three-Dimensional In Vitro Maturation (3D-IVM)

Selected COC-microbeads were rinsed with fresh IVM medium, placed in four-well Nunc dishes (Nunc Intermed, Roskilde, Denmark) containing 400 µL of IVM medium in each well, covered with an equal volume of pre-equilibrated lightweight paraffin oil (SAGE Cooper Surgical, Malov Denmark, code n. ART-4008-5) and cultured in vitro for 22–24 h at 38.5 °C under 5% CO_2_ in air. COCs not subjected to microencapsulation were cultured under the same conditions as conventional two-dimensional (2D) controls. In both 3D and 2D conditions, the mean number of COCs/well was 20–25. After IVM culture, sodium alginate microbeads were dissolved by chelation of the calcium with 2% *w*/*v* sodium citrate in IVM medium, and COCs were recovered and destined either to nuclear chromatin and bioenergetic/oxidative status analysis or to in vitro embryo production. Oocytes used for chromatin and bioenergetic status estimation, underwent CC removal by incubation in TCM-199 with 20% FCS containing 80 IU hyaluronidase/mL and aspiration in and out of finely drawn glass pipettes. Those for in vitro embryo production underwent in vitro fertilization (IVF) and in vitro embryo culture (IVEC), as described below.

### 2.6. In Vitro Fertilization (IVF) and In Vitro Embryo Culture (IVEC)

In vitro fertilization was performed in synthetic oviductal fluid medium (SOFM) [40] supplemented with 2% estrous sheep serum and 1 µg/mL heparin, as described by Martino et al. [41]. Frozen-thawed spermatozoa were selected via the swim-up technique and used at a final concentration of 1.5 × 10^6^ spermatozoa/mL. Oocytes and sperm cells were co-incubated for 22 h at 38.5 °C and under 5% CO_2_ in air in four-well dishes. After IVF culture, presumptive zygotes were freed of CCs by mechanical removal, with aspiration in and out of finely drawn glass pipettes, and cultured for 7 days in four-well dishes in SOFM with essential and non-essential amino acids at oviductal concentrations [42] and 0.4% bovine serum albumin (BSA) under mineral oil, in a maximum humidified atmosphere with 5% CO_2_, 5% O_2_, and 90% N_2_ at 38.5 °C. Blastocyst formation was assessed at day 7, and blastocysts were classified according to their degree of expansion and hatching status [41]: early blastocyst (normal blastocyst with a blastocoel equal or up to half of the embryo volume), expanded blastocyst (blastocyst with a blastocoel greater than half of the embryo volume), and hatching blastocyst (hatching or already hatched blastocyst). Moreover, blastocyst diameter was evaluated using the Oosight™ Imaging System 4.0 version (Research Instruments Ltd., Bickland Industrial Park, Falmouth, UK) as the distance between the outside borders of the trophectoderm. Live images were analyzed using the Nikon Instruments NIS-Elements BR 4.50.00 Imaging software (Mario Lippolis Strumentazione Scientifica, Bari, Italy). Then, blastocysts underwent a triple staining procedure for joint evaluation of the number of healthy or apoptotic nuclei, mitochondrial activity, and ROS levels, as described below.

### 2.7. Three-Dimensional Millifluidic In Vitro Maturation (3D mIVM)

For 3D millifluidic IVM culture (3D mIVM), the commercial LiveBox1 bioreactor (IVTech, Massarosa, Italy) was used. LiveBox1 consists of a polydimethylsiloxane (PDMS) chamber with a transparent top and bottom, designed to reproduce the typical volume of the single well of a 24-well plate (2 mL) with a flow inlet and an outlet for the perfusion of culture media. Flow lines were tangential to COC-microbeads floating in suspension but lying on the basal surface of the chamber. To create a millifluidic circuit, the culture chamber was connected with a mixing chamber, a reservoir of 10 mL of culture IVM medium, and a peristaltic pump, permitting the circulation of the culture medium from the mixing chamber to the cell chamber. Microbeads were placed inside the cell chamber after adding 300 µL of IVM medium. The chamber was then hermetically sealed and completely filled with an additional 1700 µL of IVM medium through the inlet tube. A silicone tube connected to a 0.22 µm filter allowed air flow in the chamber. The chamber was then connected to the reservoir and the peristaltic pump. To fill the circuit, the flow rate was set at 450 μL/min, and then regulated to 50 µL/min once the bioreactor was completely filled. The whole circuit was placed in a cell culture incubator, under the conditions described above. Each LiveBox1 bioreactor contained between 20 and 35 COCs/well. After IVM culture, the microbeads were dissolved and the COCs recovered, as described above. Figure 1 shows the SpHyGa bioprinter setup and the assembly of the millifluidic dynamic circuit.

### 2.8. Computational Models

To probe the effects of flow on in vitro COC cultures, computational models of the bioreactor and static culture well with the beads were implemented in COMSOL Multiphysics (Stockholm, Sweden), coupling incompressible Navier–Stokes and convection–diffusion equations [43,44]. The COC-alginate beads were modeled as concentric spheres, with an inner-cell-filled radius of 0.3 mm and an outer alginate shell of 0.7 mm thickness, placed at the bottom of the bioreactor (bead number = 21). The fluid viscosity of the culture medium was assumed to be equal to that of water at 37 °C (i.e., 0.6913 mPa/s), while the oxygen diffusion coefficient was assumed to be equal to 3 × 10^−9^ in the media and 2.5 × 10^−9^ m^2^/s in the gels [45]. COC oxygen consumption rate (OCR = 4 pmol/min) was estimated regarding data reported in the literature for bovine COCs [46]. For IVM studies, performed under 5% CO_2_ in air, the initial oxygen concentration and the oxygen partial pressure were set at 0.21 mol/m^3^ and 159 mmHg, respectively. An exploratory computational study was also implemented in order to assess the possibility of using millifluidic bioreactors for IVF cultures, for which some studies support the use of hypoxic conditions [47,48,49]. In this case, considering an incubator atmosphere with 5% CO_2_, 5% O_2_, and 90% N_2_, the initial oxygen concentration and the oxygen partial pressure were set at 0.05 mol/m^3^ and 38 mmHg, respectively. The bioreactor was modeled as a cylindrical chamber with PDMS lateral walls (oxygen permeability = 3.78 × 10^−11^ mol m m^−2^ s^−1^ mmHg) [43,44], while the static well was modeled as a cylindrical geometry with a media layer of 2.1 mm covered by an oil layer of 2.1 mm with an oxygen diffusion coefficient of 2 × 10^−9^ m^2^/s [50].

### 2.9. Granulosa Cell Isolation

For GC isolation, prepubertal lamb ovaries were washed in sterile PBS and subjected to the slicing procedure, as described above. Collected fluids were observed under a Nikon SMZ18 stereoscopic microscope, and GCs were morphologically identified and subsequently removed using a Gilson pipette with sterile tips. GC suspensions were pooled and centrifuged at 600× *g* for 10 min [51]. Cell pellets were washed 2–3 times in MPM with 20% FCS and then resuspended in IVM medium. The number of living cells was estimated using a Scepter 2.0 handheld automated cell counter (Millipore Corporation, Billerica, MA, USA) equipped with 60 µm sensors.

### 2.10. Type I Collagen Preparation

Type I collagen 0.4% (CI; Sigma-Aldrich C-3867 from rat tail) was used by dissolving 100 mg of proteins in 25 mL of 20 mmol/L acetic acid. This solution was diluted (1/100) to a working concentration of 0.004% with sterile tissue-culture-grade water and used for coating wells or microbead preparation, as described below.

### 2.11. COC-GC Co-Cultures

For four-well dish coating, 200 μL of CI working solution was added to each well, and the protein was allowed to bind for 3 hours on a heating stage set at 37 °C. The excess fluid was removed from the coated surfaces and allowed to dry. Freshly isolated GCs in IVM medium were seeded at a concentration of 10^6^ viable cells/mL. As cell suspension volume varied according to cell concentration (around 100–130 µL), the final volume of 400 µL in each well was adjusted with additional IVM medium. An equal volume of mineral oil overlay, equilibrated under 5% CO_2_ in air, was added, and COCs or COC-microbeads were placed and co-cultured with GCs at 38.5 °C under 5% CO_2_ in air. Uncoated wells, with or without GCs, were used as controls.

For microbead preparation, 0.4% CI solution was added to alginate solution in order to attain a final CI concentration of 0.004%. GCs were suspended in this solution at a concentration of 10^6^ viable cells/mL. The solution was uploaded to the SpHyGa syringe to be used for microbead preparation. Microbeads without CI and/or GCs were used as controls. The cumulus cell apoptotic index, as related to GC and CI addition, was analyzed by morphological assessment, as reported by Mastrorocco et al. [52].

### 2.12. Oocyte and Blastocyst Mitochondria and ROS Staining

Oocytes and blastocysts were washed three times in PBS with 3% BSA and incubated for 30 min in the same medium containing 280 nmol/L MitoTracker Orange CMTMRos (Molecular Probes) at 38.5 °C under 5% CO_2_ in air [41,53]. Negative controls were analyzed after staining with the MitoTracker Orange and further incubation for 5 min in the presence of 5 µmol/L of the mitochondrial membrane potential (delta psi)-collapsing uncoupler carbonyl cyanide 3-chlorophenylhydrazone (CCCP; Molecular Probes) [52]. After incubation with MitoTracker, oocytes and blastocysts were washed in PBS with 0.3% BSA and incubated for 15 min at 38.5 °C under 5% CO_2,_ in air, in the same medium containing 10 µmol/L 2′,7′-dichlorodihydrofluorescin diacetate (H_2_DCF-DA) in order to detect the dichlorofluorescein (DCF) and localize intracellular sources of ROS [54]. After incubation, oocytes and blastocysts were washed in PBS without BSA and fixed overnight at 4 °C in 2% paraformaldehyde solution in PBS [52]. Particular attention was applied to avoid sample exposure to the light during staining and fixing procedures and reduce photobleaching.

### 2.13. Nuclear Chromatin Evaluation of Oocytes and Embryos

To evaluate oocyte and embryo nuclear chromatin, after the fixation in 2% paraformaldehyde solution in PBS, oocytes and embryos were stained with 2.5 μg/mL Hoechst 33258 in 3:1 (*v*/*v*) glycerol/PBS mounted on microscope slides with coverslips, sealed with nail polish, and kept at 4 °C in the dark until observation. Slides were examined under an epifluorescence microscope (Nikon Eclipse 600; ×400 magnification) equipped with a B-2A (346 nm excitation/460 nm emission) filter. Oocytes were evaluated in relation to their meiotic stage, and classified as germinal vesicles (GVs), metaphase to telophase I (MI to TI), and MII with the first polar body extruded [41]. Oocytes showing either multipolar meiotic spindle, irregular chromatin clumps, or absence of chromatin were considered to be abnormal [36]. Oocytes with the first polar body (PB) extruded and one pronucleus were classified as parthenogenetically activated [55]. Embryos were classified according to their number of nuclei. They were indicated as: (1) normal, when the presence of a regular-shaped nucleus inside each blastomere was observed; (2) morulae, if they contained more than 32 cells but did not have an organized outer layer of cells; or (3) blastocysts, if they contained more than 64 cells and had begun the organization of outer presumptive trophoblast cells. The formation of micronuclei and lobulated nuclei was considered to be a sign of chromatin damage [53].

### 2.14. Assessment of Mitochondrial Distribution Pattern and Intracellular ROS Localization

Oocytes, at the MII stage, and blastocysts were observed at ×600 magnification in oil immersion with a Nikon C1/TE2000-U laser scanning confocal microscope. A 543 nm helium/neon laser and the G-2A filter were used to detect the MitoTracker Orange CMTMRos (551 nm excitation and 576 nm emission). A 488 nm argon-ion laser and the B-2A filter were used to detect DCF (495 nm excitation and 519 nm emission). Scanning was conducted with 25 optical sections from the top to the bottom of the oocytes and embryos, with a step size of 0.45 μm to allow for 3D distribution analysis. The mitochondrial distribution pattern was evaluated on the basis of previous studies [41,53,56]. Briefly, (1) a homogeneous distribution of small mitochondria aggregates throughout the cytoplasm was considered to be an indication of a low-energy cytoplasmic condition; (2) a perinuclear and subplasmalemmal (P/S) distribution of mitochondria forming large granules was considered to be characteristic of healthy cytoplasmic conditions; and (3) an irregular distribution of mitochondria, with large mitochondrial clusters, were classified as abnormal. Concerning intracellular ROS localization, oocytes and embryos with intracellular ROS distributed throughout the cytoplasm, together with areas/sites of mitochondria/ROS overlapping, were considered to be healthy [41,53,56].

### 2.15. Quantification of MitoTracker Orange CMTMRos and H_2_DCF-DA Fluorescence Intensity

In each individual MII oocyte, MitoTracker and DCF fluorescence intensities were measured at the equatorial plane, at the excitation/emission as described above by use of EZ-C1 Gold Version 3.70 image analysis software platform for the Nikon C1 confocal microscope. A circular area was drawn in order to measure only the region including cell cytoplasm. For the blastocysts, a circle was traced on all the 25 acquired focal planes. The fluorescence intensity within the programmed scan area was recorded and plotted against the conventional pixel unit scale (0–2.16255). Mitochondrial activity and intracellular ROS levels were recorded as MitoTracker Orange CMTMRos and DFC fluorescence intensity in arbitrary densitometric units (ADU). Parameters related to fluorescence intensity, such as laser energy, signal detection (gain), and pinhole size, were maintained at constant values for all measurements.

### 2.16. Mitochondria/ROS Colocalization Analysis

Colocalization analysis of mitochondria and ROS was performed with the EZ-C1 Gold software (version 3.70). The degree of colocalization was reported with correlation coefficients quantifying the overlap degree between the MitoTracker Orange CMTMRos and DCF fluorescence signals. Mitochondria/ROS colocalization has been reported as a biomarker of healthy oocytes and embryos [41,52,53,56,57].

### 2.17. Statistical Analysis

The proportions of oocytes showing different chromatin configurations and mitochondrial distribution patterns, cleaved embryos, and blastocysts were compared between groups using the chi-squared test. Blastocyst diameter, total number of nuclei, and percentage of apoptotic nuclei were compared using Student’s unpaired *t*-test. For mitochondria and ROS quantification analysis, data (mean ± standard deviation (SD)) were expressed as a percentage of the control sample signal and compared using Student’s *t*-test or one-way ANOVA, followed by Tukey’s post-hoc multiple comparison test. Differences with *p* < 0.05 were considered to be statistically significant.

## 3. Results

### 3.1. 3D-IVM Improved Prepubertal Oocyte Maturation Rate and Bioenergetic/Oxidative Status

The effects of the 3D-IVM on oocyte maturation rate and bioenergetic/oxidative status were evaluated by 8 independent runs in which 353 COCs were processed. In both 3D and 2D conditions, cumulus integrity and oocyte structure were preserved. In 3D-IVM, the percentage of oocytes that reached the MII stage was significantly increased compared with 2D controls (*p* < 0.01, Table 1). Correspondingly, in the 3D-IVM system, a significantly reduced percentage of oocytes showing abnormal chromatin configurations was observed (*p* < 0.05, Table 1).

Bioenergetic/redox status evaluation of matured oocytes showed that 3D-IVM improved oocyte mitochondrial distribution patterns. Indeed, the percentage of oocytes showing heterogeneous perinuclear and subcortical mitochondrial distribution (P/S), indicating cytoplasm maturity and competence, significantly increased in the group of MII oocytes derived from 3D-IVM compared with 2D (*p* < 0.05; Table 1).

Oocyte mitochondrial membrane potential and intracellular ROS levels significantly increased after 3D-IVM compared with 2D controls, as assessed by increased MitoTracker Orange CMTMRos (*p* < 0.001) and DCF (*p* < 0.01) fluorescence intensity. Mitochondrial/ROS colocalization did not vary between the two IVM conditions. Figure 2 shows quantification data of the bioenergetic/oxidative parameters of MII oocytes cultured in 2D-versus 3D-IVM.

### 3.2. 3D-IVM Improved Prepubertal Embryo Cleavage and Blastocyst Quality

Additional independent IVM experiments in 3D versus 2D conditions, followed by IVF and IVC, showed that after oocyte culture in 3D-IVM the total cleavage increased significantly compared with controls (*p* < 0.05; Table 2).

Although no effects were observed on the blastocyst formation rates, blastocyst viability was improved after oocyte culture in 3D-IVM. Indeed, blastocysts derived from 3D-IVM tended, without attaining statistical significance, to display increased diameter (Figure 3A) and a higher number of nuclei (Figure 3B), and showed a significantly lower percentage of cells with apoptotic chromatin (*p* < 0.05, Figure 3C). On the other hand, blastocyst distribution among early, expanded, and hatched stages did not vary according to oocyte IVM condition.

Blastocyst mitochondrial membrane potential and mitochondrial/ROS colocalization significantly increased after 3D-IVM compared with 2D controls (*p* < 0.01 and *p* < 0.001, respectively; Figure 4). Instead, intracellular ROS levels were significantly reduced in blastocysts obtained from 3D-IVM-matured oocytes (*p* < 0.01; Figure 4).

### 3.3. 3D Millifluidic IVM (3D-mIVM) Boosted Oocyte Bioenergetic/Oxidative Status

The effects of 3D-mIVM on oocyte maturation rate and bioenergetic/oxidative status, were analyzed in 6–8 independent runs, in which 357 COCs were processed. Oocyte nuclear maturation rate was not affected by the 3D-mIVM system, nor was the mitochondrial distribution pattern (differences statistically insignificant, Table 3).

Interestingly, oocyte mitochondrial membrane potential, intracellular ROS levels, and mitochondria/ROS colocalization were significantly increased after 3D-mIVM, compared with 3D-IVM controls, as assessed by increased MitoTracker Orange CMTMRos (*p* < 0.001) and DCF (*p* < 0.001) fluorescence intensity and overlap coefficient (*p* < 0.05; Figure 5).

### 3.4. Shear Stress and Nutrient Supply Were Enhanced with Flow

Computed shear stress distribution at the bottom of the bioreactor (without COC-microbeads) for a flow rate of 50 μL/min shows a maximum shear stress value (1.2 × 10^−6^ Pa) at the center, which decreased toward the edges (Figure 6A). When placed in the LiveBox1 system, the shear stress calculated on the surface of the alginate beads was slightly higher for those in the center (3.7 ± 0.3 × 10^−5^ Pa) compared to those at the edges (1.2 ± 0.1 × 10^−5^ Pa).

The convection–diffusion–reaction model (Figure 6B,I) demonstrated that millifluidic conditions improved oxygen distribution as, in the bioreactor, oxygen levels were consistently higher than in the well, in both normoxic (Figure 6B vs. Figure 6C) and hypoxic (Figure 6F vs. Figure 6G) conditions. This is due to the flow and the oxygen permeability of PDMS [42,43].

The oxygen profiles of one of the centrally positioned COC-microbeads in the bioreactor and the well are shown for normoxic (Figure 6D,E) and hypoxic (Figure 6H,I) conditions, respectively. Oxygen diffuses through the alginate shell and is consumed in the COC core. The mean oxygen concentration in the beads was calculated as the integral of the oxygen concentration over bead volume. In normoxic conditions (for IVM), cell oxygen consumption did not significantly affect oxygen distribution. The mean oxygen concentration in the beads was higher in the bioreactor (0.197 mol/m^3^) with respect to the static well (0.184 mol/m^3^). Similarly, in hypoxic conditions (for IVF), the mean oxygen concentration was equal to 0.040 mol/m^3^ in the bioreactor and 0.032 mol/m^3^ in the well.

### 3.5. Granulosa Cells and Type I Collagen Improved Oocyte Maturation and Bioenergetic/Oxidative Status

Granulosa cells did not improve oocyte nuclear maturation rates when seeded onto uncoated tissue culture plastic wells (Table 4, lane 1 vs. lane 2, differences statistically insignificant). Moreover, GCs’ addition increased the rate of oocytes that did not resume meiosis and remained blocked at the GV stage (*p* < 0.05). When added in microbeads, GC addition worsened oocyte maturation (lane 3 vs. lane 4, *p* < 0.001). This inhibition was found to be associated with significant increase of GV-stage (*p* < 0.01) and abnormal oocytes (*p* < 0.001), and reduction of MI/TI oocytes (*p* < 0.01). Notably, the percentage of MII oocytes obtained after 3D-IVM was higher compared to controls in 2D-IVM (*p* < 0.05), and the percentage of abnormal oocytes was significantly reduced (*p* < 0.001), confirming the results of our previous study [35].

Concerning cytoplasmic maturation, GC addition was ineffective in both 2D and 3D conditions (lane 1 vs. lane 2, and lane 3 vs. lane 4; differences statistically insignificant). However, in oocytes matured under 3D-IVM conditions, the percentage of those showing P/S mitochondrial distribution patterns, indicating ooplasmic maturity, was significantly higher than that of oocytes matured under 2D conditions (overall data: 56/118, 47% versus 29/109, 27%; *p* < 0.05; Table 4), thus confirming the results of previous studies [35].

In the absence of GCs, CI addition was not effective in terms of nuclear maturation (Table 4, lane 3 vs. 5, differences statistically insignificant). Conversely, when CI was added in the presence of GCs, it significantly improved oocyte nuclear maturation, either in microbeads (lane 4 vs. 6: *p* < 0.01) or as a plate coating (lane 4 vs. 7: *p* < 0.01). This result was found to be associated with significant reduction of the percentages of oocytes with abnormal chromatin configuration (*p* < 0.01). Interestingly, maturation rate variations corresponded to alteration of cumulus cell apoptotic indices.

Oocytes matured in microbeads without GCs, and with or without CI, showed unchanged mitochondrial patterns (Table 4, lane 3 vs. 5; differences statistically insignificant). When oocytes were cultured with GCs and with CI, they showed significantly increased percentages of P/S mitochondrial distribution patterns only when CI was included in microbeads (*p* < 0.05) while no increase was observed with CI as a plate coating.

Concerning quantification bioenergetics analysis, in oocytes matured in COC-microbeads with GCs and CI included, mitochondrial activity significantly increased (*p* < 0.001), intracellular ROS remained unvaried, and mitochondria/ROS colocalization increased (*p* < 0.01) compared with those cultured with GCs and without CI; instead, in oocytes matured in COC-microbeads cultured on a GC monolayer coated with CI, all quantification parameters remained unchanged compared with those cultured without CI (Figure 7).

Concerning GCs’ morphology, when cultured on uncoated tissue culture plastic wells, they showed homogeneous, fibroblast-like morphology, whereas after 24 h culture in CI-coated wells, they appeared as a heterogeneous mix of fibroblast-like and round-shaped cells, rather flat in both cases. When included in microbeads, they showed a spherical shape, with some cytoplasmic protrusions, either in the presence or the absence of CI (Figure 8).

Oocytes issuing from the examined bioengineering approaches varied in their mitochondrial patterns and activity and ROS levels (Figure 9).

## 4. Discussion

Although bioengineering strategies are becoming widespread in the area of cell culture, including ARTs, to the best of our knowledge such approaches have never been applied in prepubertal oocytes in studies conducted to date, but only in adult oocytes [36,51,58,59,60,61,62].

The aim of the present study was to mimic as much as possible the physiological follicular conditions, such as COC microencapsulation, for 3D-IVM. The first step was to provide prepubertal COCs with a 3D environment using a one-step automated bioprinting method [36]. As the results show, the applied technique was able to improve prepubertal oocyte nuclear maturation rates and to reduce the incidence of chromosomal abnormalities. In addition, a higher percentage of oocytes matured under 3D-IVM showed localized redistribution of mitochondria in P/S areas. This feature is functional to the acquisition of nuclear/cytoplasmic maturation and embryo developmental potential [63]. Moreover, oocytes matured under 3D-IVM showed increased mitochondrial activity and corresponding ROS generation ability, with unaltered colocalization of intracellular ROS with active mitochondria. These results confirm the results obtained previously with adult sheep oocytes [36]. As a biomarker of cytoplasmic condition, these analyzed parameters highlighted that, after 3D-IVM, oocytes from prepubertal lambs have a well-balanced redox status, as they appear to be healthy and non-oxidized. The presence of active mitochondria and higher ATP levels in oocytes correlate with better embryo development and implantation rates [64]. As long-term effects, 3D conditions were found to have a good impact on blastocyst quality, since blastocysts derived from 3D-IVM oocytes showed significantly increased mitochondrial membrane potential and significantly reduced apoptotic index and ROS levels. Increased mitochondrial membrane potential indicates that these blastocysts exhibit improved mitochondrial activity; the increased mitochondria/ROS colocalization associated with a reduction in ROS levels is indicative of a well-balanced oxidative state, as produced ROS are mostly located in the mitochondria, where they are physiologically produced, instead of being poured into non-physiological cytoplasmic areas, where they can be a source of cell damage. Therefore, 3D-IVM improved oocyte quality, with positive effects on blastocysts’ quality. To the best of our knowledge, this is the first study in which prepubertal COCs matured under 3D-IVM were shown to produce viable embryos in vitro. To date, only one study on mice has reported normal embryonic and fetal development with the birth of live, healthy, and fertile male and female offspring by using a tissue engineering approach [65]. In that study, follicles from prepubertal mice were encapsulated in alginate microbeads, grown in vitro for 8 days and, subsequently, the isolated oocytes underwent IVF, IVC, and successful implantation into host mothers [65].

In recent years, cell culture bioreactors have been developed to overcome the limits of static cell cultures and to allow in vitro cultures under physiological flow conditions [66]. In the present study, the LiveBox1 bioreactor was used to provide the cells with in-vivo-like shear stress and better nutrient turnover and catabolite removal [67,68]. To the best of our knowledge, this is the first study to introduce a millifluidic approach in 3D-IVM applications for a large animal species. Oocytes matured in the bioreactor showed improved bioenergetic status compared to controls, suggesting that nutrient supply was improved compared to static conditions. This was also confirmed by the computational models, which showed optimal oxygen concentrations for all encapsulated COCs, which were consistently higher than those in the wells in both normoxic and hypoxic conditions. This suggests that the bioreactors can be also used for further IVF studies.

Our models also showed that the shear stress on the beads was very low, particularly for those beads located at the periphery of the bioreactor. In fact, maturation rates between oocytes in the bioreactor chamber versus static controls were similar, consistent with the results of Ledda et al. [69]. However, despite the lack of any effects on nuclear maturation, the improved cytoplasmic maturation, observed in terms of enhanced bioenergetics, can be considered to be a crucial biomarker of oocyte developmental potential. Therefore, further studies are needed in order to investigate higher flow rates, which result in higher shear stresses [67], or different bead positioning to optimize shear stress levels. Moreover, as the LiveBox1 platform allows sequential connection of two or more cell lines [68], future studies could concern the effects of different cells of the female reproductive tract on COC competence.

In the last approach, the effects of adding GCs in co-culture with COCs—thus, recreating the physiological 3D follicular environment—were analyzed [70,71,72,73,74,75]. Granulosa cells, seeded onto uncoated tissue culture plastic wells, adhered to the bottom of the well and flattened in monolayers, displaying a fibroblast-like morphology and behavior [72,73]. Conversely, when plated on CI-based coatings or gels, GCs maintained their spherical, epithelioid shape, and showed estrogenic secretory activity [76,77,78]. Results obtained in COC–GC co-cultures performed in 2D conditions confirm the previous observations. Oocyte maturation and bioenergetics were not enhanced upon GCs’ addition, and GCs showed the expected fibroblast-like morphology, allowing us to hypothesize progesterone secretory activity [78]. Granulosa cells included in alginate-based COC-microbeads maintained their spherical native phenotype. However, no ameliorative effects on oocytes were noticed. Rather, multiple damaging effects were found, such as reduction in oocyte nuclear maturation with a corresponding increase in the percentage of oocytes with abnormal chromatin configurations, or that remained blocked at the GV stage. Moreover, matured oocytes did not show a healthy mitochondrial distribution pattern. This could be related to alginate’s properties, such as the absence of cell adhesive sequences or its hydrophilic nature [77,79].

Conversely, when GCs were included in alginate CI COC-microbeads (3D + iCI + iGCs) or on CI-coated plastic wells, the oocyte maturation rate was significantly increased. Moreover, oocytes matured in COC-microbeads with GCs in the presence of CI also showed an improved bioenergetic status. In these particular microbeads, GCs maintained their spherical epithelioid shape, while when cultured on the CI-coated layer, a mix of GCs with spherical and fibroblast-like morphology was found. These results confirm the observation that the culture of GCs under 3D conditions and in the presence of CI contributes to the maintenance of native cell phenotypes, presumably directing them into the estradiol-secreting pathway, as well as proliferation and survival [76,77]. By way of confirmation, microbeads fabricated as a hydrogel network of alginate and CI without GCs did not improve oocyte maturation, indicating the importance of GC presence.

In conclusion, the bioengineering approaches used in this study improved the in vitro performance of prepubertal oocytes. Possible synergic uses could bring improvements to JIVET.

## Figures and Tables

**Figure 1 cells-10-01458-f001:**
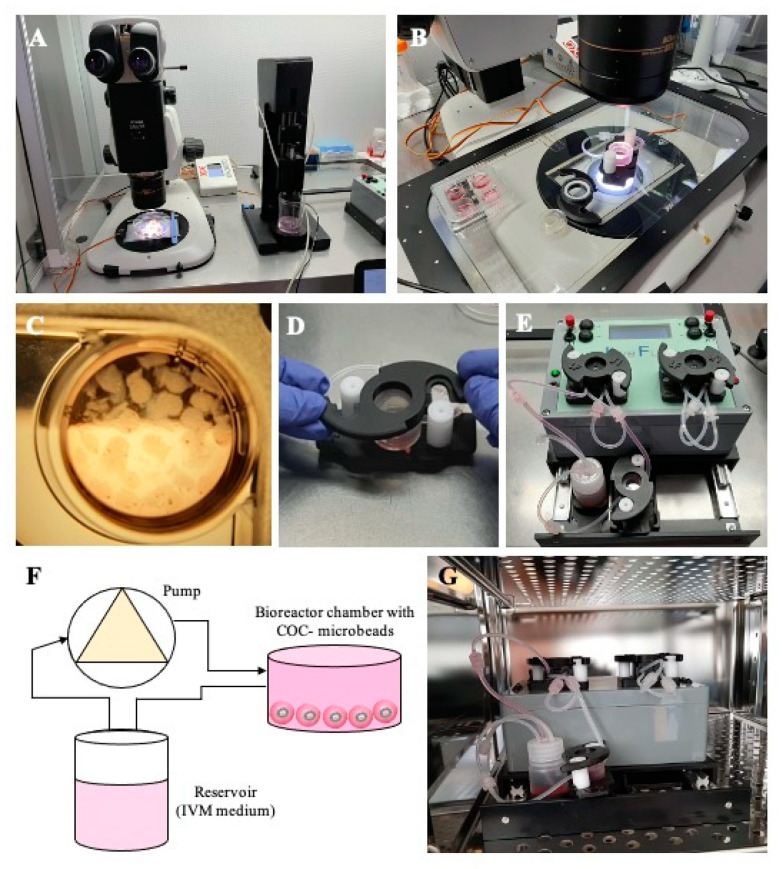
Set up of 3D millifluidic IVM. (**A**) Spherical hydrogel generator (SpHyGa) set up for the preparation procedure of COC-microbeads of prepubertal lambs; stereomicroscope for slicing of prepubertal lamb ovaries and COC collection, and SpHyGa extrusion module with dedicated software, both under a laminar flow hood. (**B**) A four-well plate and a LiveBox1 chamber on a heating stage and under a stereomicroscope, ready to be used for COC-microbead placement. (**C**) A well with COC-microbeads for 3D-IVM. (**D**) LiveBox1 culture chamber assembly. (**E**,**F**) Millifluidic circuit composed of a LiveBox1 chamber, reservoir with IVM medium, and peristaltic pump. (**G**) LiveBox1 placed in the incubator under 5% CO_2_ in air.

**Figure 2 cells-10-01458-f002:**
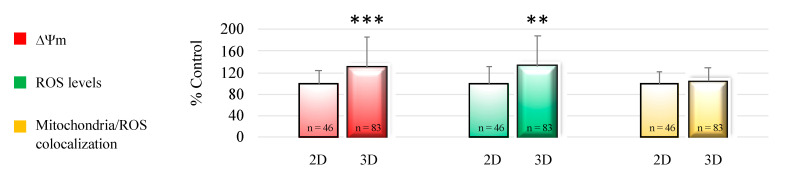
Effects of 3D-IVM on prepubertal lamb oocyte bioenergetic/oxidative status. Quantification data of bioenergetic/oxidative parameters of MII oocytes cultured in 2D- versus 3D-IVM. Mitochondrial activity (ΔΨm) and intracellular ROS levels are expressed as a percentage of the signal of the control samples (oocytes cultured in 2D-IVM). Mitochondria/ROS colocalization is expressed as a percentage of the correlation coefficient value of the control samples. Numbers of analyzed matured oocytes, derived from six IVM runs per condition, are indicated at the bottom of each graph bar. For each oocyte, data were obtained at the equatorial plane. Student’s *t*-test: ** *p* < 0.01; *** *p* < 0.001.

**Figure 3 cells-10-01458-f003:**
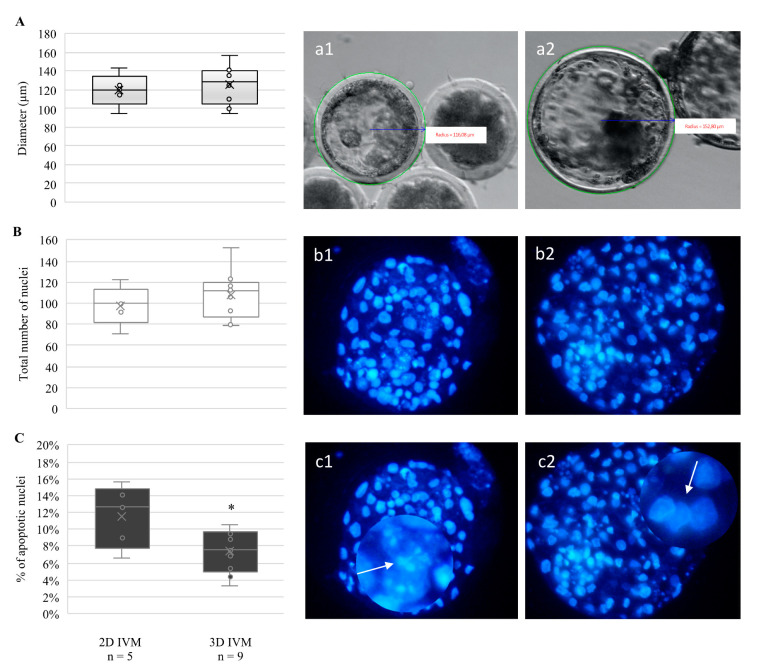
Effects of 3D-IVM on the quality of blastocysts obtained from prepubertal lamb oocytes. (**A**) Boxplots representing blastocyst diameters. (**B**) Boxplots representing the total number of nuclei per blastocyst. (**C**) Boxplots representing the percentage of apoptotic nuclei per blastocyst. Student’s *t*-test comparisons of 2D-IVM vs. 3D-IVM; * *p* < 0.05. (**a1**,**a2**) Blastocyst diameters and (**b1**,**b2**) nuclear chromatin after Hoechst 33258 staining. (**c1**,**c2**) Apoptotic and healthy nuclei, respectively, can be seen as enlarged details.

**Figure 4 cells-10-01458-f004:**
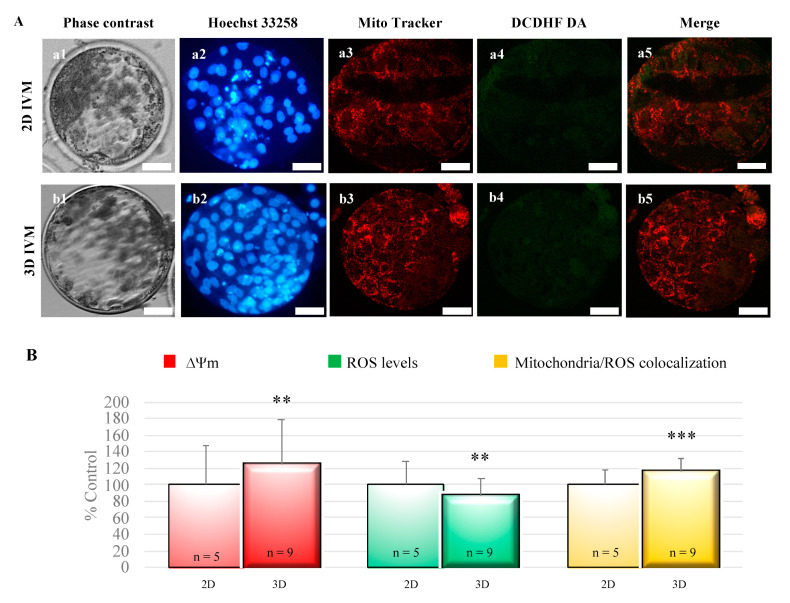
Effects of 3D-IVM on blastocyst bioenergetic/oxidative status. (**A**) Photomicrographs showing representative images of blastocysts derived from oocytes subjected to 2D- (lane a) or 3D-IVM (lane b). MitoTracker Orange and DCDH FDA were used to label mitochondria and ROS, respectively. Nuclear chromatin was stained with Hoechst 33258. For each embryo, the corresponding phase contrast (column 1), epifluorescence images showing nuclear chromatin (column 2), and confocal images showing the mitochondrial distribution pattern (column 3), ROS localization (column 4), and mitochondria/ROS merge (column 5) are shown. The 2D blastocyst was expanded, and shows a dark area indicating the blastocoelic cavity, whereas the 3D blastocyst hatched, and shows groups of cells protruding through the zona pellucida. Scale bars represent 40 μm. (**B**) Quantification data of bioenergetic/oxidative parameters of blastocysts obtained from oocytes cultured in 2D- versus 3D-IVM. Mitochondrial activity (ΔΨm) and intracellular ROS levels are expressed as percentages of the signal value of the control samples (blastocysts obtained from oocytes cultured in 2D-IVM). Mitochondria/ROS colocalization is expressed as a percentage of the correlation coefficient value of the control samples. Numbers of analyzed blastocysts are indicated at the bottom of each graph bar. For each blastocyst, data were obtained at 25 serial scans from the top to the bottom. Student’s *t*-test: ** *p* < 0.01; *** *p* < 0.001.

**Figure 5 cells-10-01458-f005:**
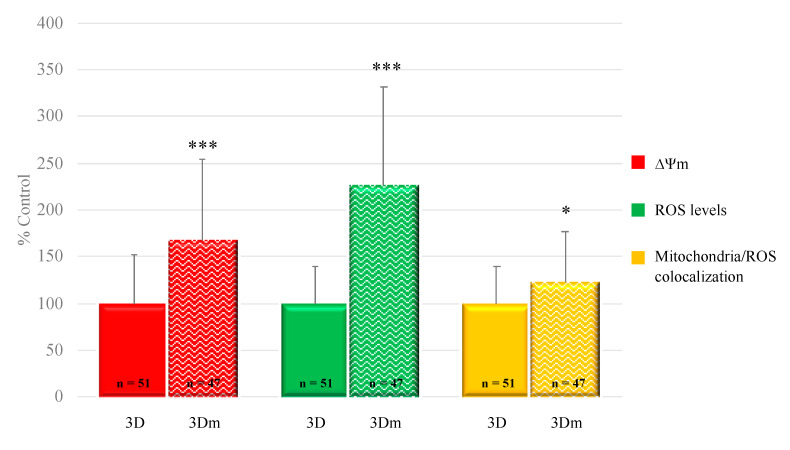
Effects of 3D-mIVM on oocyte bioenergetic/oxidative status. Quantification data of bioenergetic/oxidative parameters of MII oocytes cultured in 3D-IVM versus 3D-mIVM. Mitochondrial activity (ΔΨm) and intracellular ROS levels are expressed as percentages of the signal of the control samples (oocytes cultured in 3D-IVM). Mitochondria/ROS colocalization is expressed as a percentage of the correlation coefficient value of the control samples. Numbers of analyzed matured oocytes, derived from 3D-IVM runs per condition, are indicated at the bottom of each graph bar. For each oocyte, data were obtained at the equatorial plane. Student’s *t* test: * *p* < 0.05; *** *p* < 0.001.

**Figure 6 cells-10-01458-f006:**
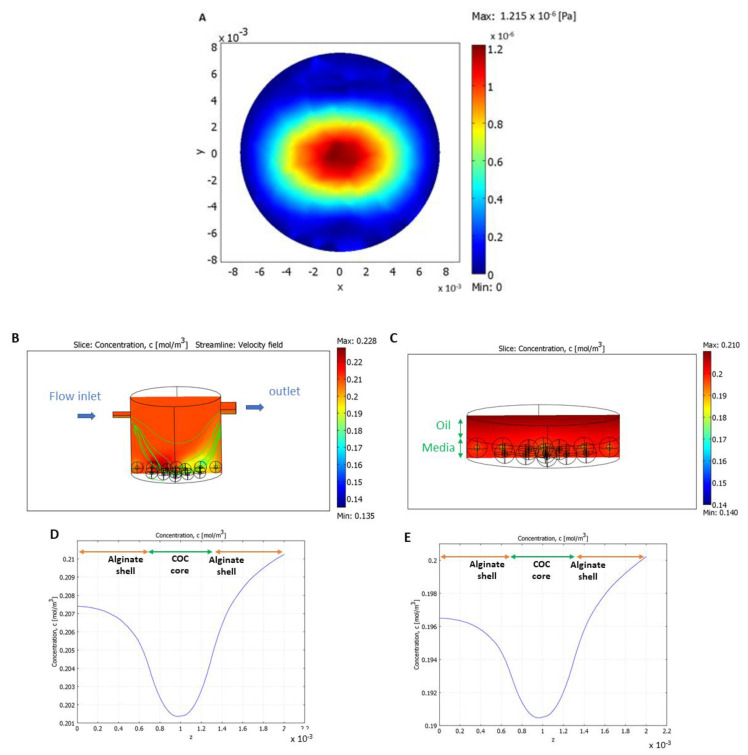
Computational models. (**A**) Shear–stress distribution without COCs at the bottom of the bioreactor; (**B**) the bioreactor showing oxygen concentration and velocity field streamlines in normoxic (5% CO_2_ in air) conditions; (**C**) a culture well showing oxygen concentration in static normoxic conditions; (**D**,**E**) oxygen profiles for alginate-COC beads placed at the center of the bioreactor and of the static well, respectively, in normoxic conditions; (**F**) the bioreactor showing oxygen concentration and velocity field streamlines in hypoxic (5% O_2_, 5% CO_2_, and 90% N_2_) conditions; (**G**) a culture well showing oxygen concentration in static hypoxic conditions; and (**H**,**I**) oxygen profiles for alginate-COC beads placed at the center of the bioreactor and of the static well, respectively in hypoxic conditions.

**Figure 7 cells-10-01458-f007:**
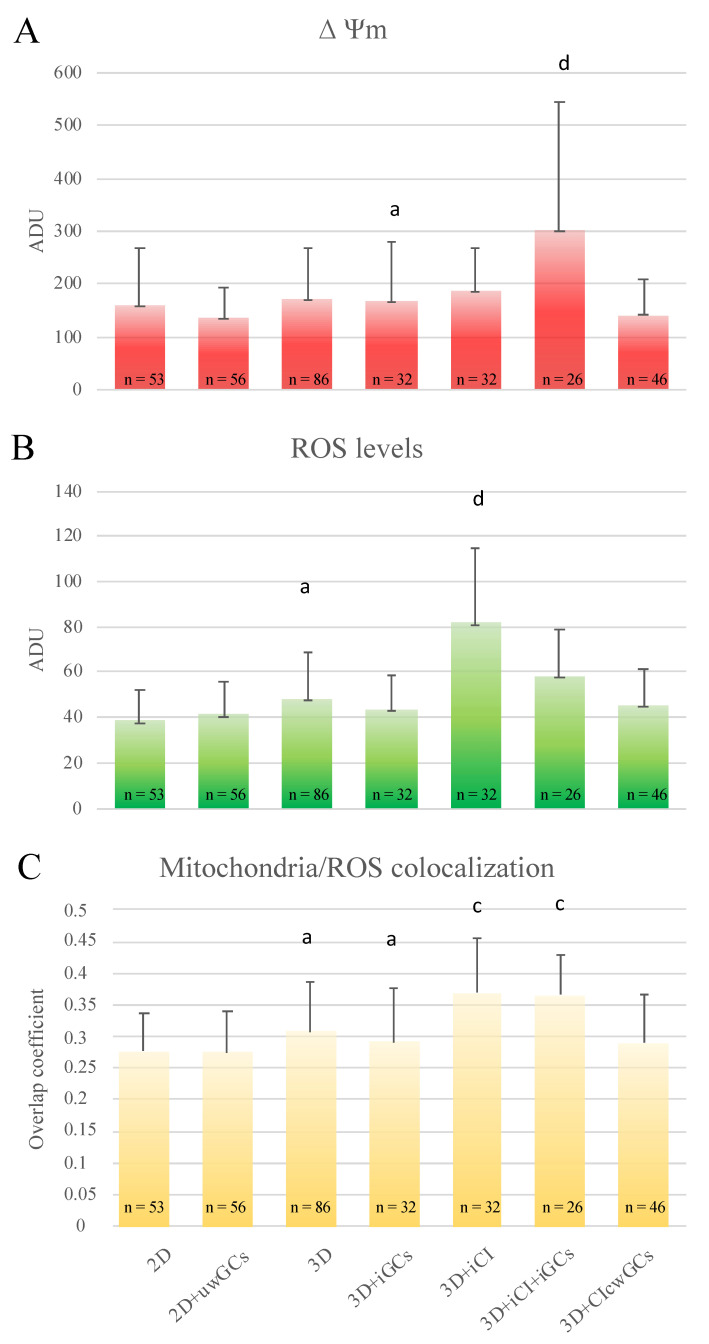
Effects of CI and GCs’presence/absence on oocyte bioenergetic/oxidative status. Quantification data of bioenergetic/oxidative parameters of MII oocytes cultured in presence/absence of CI and GCs (see Table 4). Mitochondrial activity (**A**) and intracellular ROS levels (**B**) were expressed as the means ± SD of MitoTracker Orange CMTMRos and DFC fluorescence intensity in arbitrary densitometric units (ADU). Mitochondria/ROS colocalization (**C**) was expressed as the mean ± SD of the overlap coefficient. Comparisons were performed between: 2D-IVM vs. 2D-IVM with uncoated well GCs (2D + uwGCs); 3D-IVM vs. 3D-IVM with included GCs (3D + iGCs); 3D-IVM vs. 3D-IVM with included collagen (3D + iCI); 3D-IVM with included GCs (3D + iGCs) vs. 3D-IVM with included collagen and GCs (3D + iCI + iGCs); and 3D-IVM + iGCs vs. 3D-IVM with collagen coated well GCs (3D + CIcwGCs). Due to three control groups, data were not converted into percentages of control samples. Numbers of analyzed matured oocytes are indicated at the bottom of each graph bar. For each oocyte, data were obtained at the equatorial plane. One-way ANOVA followed by Tukey’s post-hoc mul-tiple comparison test: ^a,c^ = *p* < 0.01; ^a,d^ = *p* < 0.001.

**Figure 8 cells-10-01458-f008:**
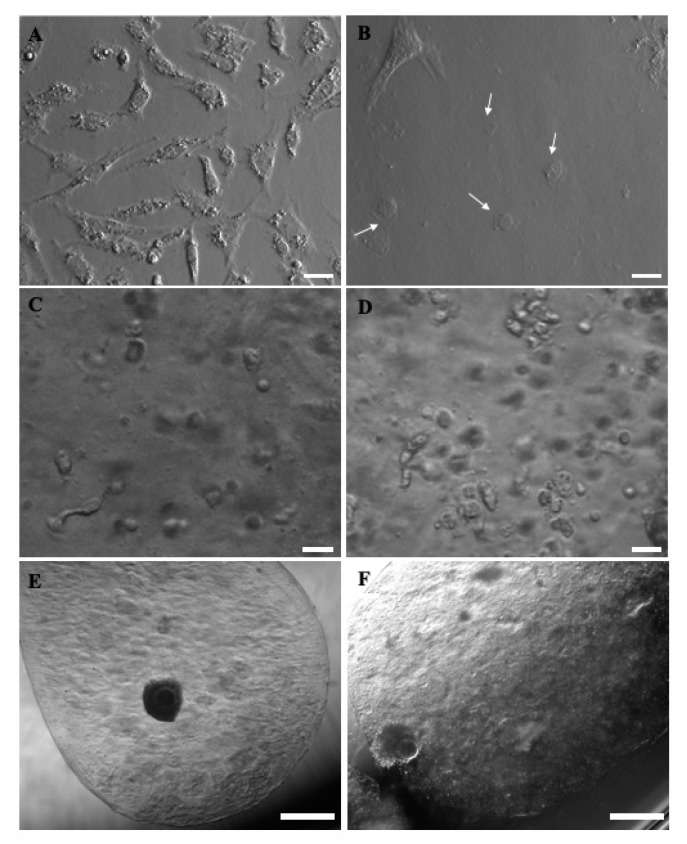
Morphology of prepubertal GCs and COCs as can be seen before or after 24 h in vitro culture in different bioengineered conditions. (**A**) GCs after 24 h culture in uncoated tissue culture plastic wells, exhibiting a homogeneous, fibroblast-like morphology. (**B**) GCs after 24 h culture in CI coated wells, displaying a heterogeneous mix of fibroblast-like and round-shaped morphology (white arrows). When included in microbeads, spherical shapes with some cytoplasmic pro Table 3. D-IVM, and (**F**) GC-including COC-microbead with an expanded COC, as observed after 3D-IVM. Scale bars represent: (**A**–**D**) 25 µm and (**E**–**F**) 500 µm.

**Figure 9 cells-10-01458-f009:**
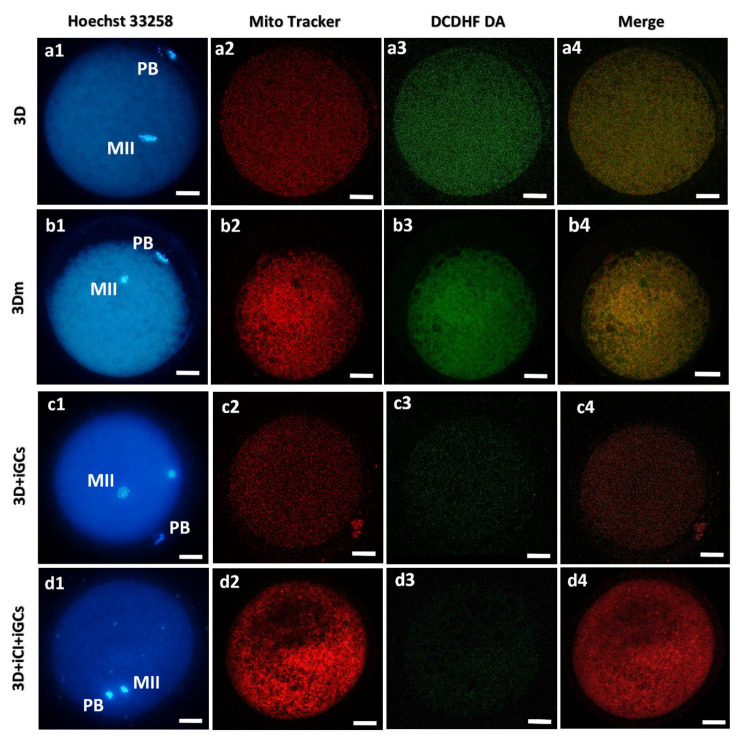
Overview of the effects of 3D-IVM, millifluidic culture, and 3D co-culture with CI and GCs on the bioenergetic parameters of mature oocytes. Photomicrographs showing representative images of prepubertal lamb oocytes after 3D-IVM (lane a), 3D-mIVM (lane b), 3D-IVM with included GCs (3D + iGCs; lane c), and 3D-IVM with included CI and included GCs (3D + iCI + iGCs; lane d). Corresponding epifluorescence images showing nuclear chromatin configuration at the MII stage (column 1: Hoechst 33258) and confocal images, taken at the oocytes’ equatorial planes, showing mitochondrial distribution patterns and activity (column 2: MitoTracker Orange CMTMRos), intracellular ROS localization and levels (column 3: DCDHF DA), and mitochondria/ROS colocalization (column 4: Merge). Images are representative of matured oocytes with homogeneous (small aggregates: a2) or heterogeneous (pericortical/perinuclear: b2, c2, d2) mitochondrial distribution patterns. All oocytes matured under 3D-IVM clearly show mitochondria distributed in the pericortical cytoplasmic area, and in the hemisphere where the meiotic spindle is located. Scale bars represent 40 μm.

**Table 1 cells-10-01458-t001:** Effects of 3D-IVM culture in alginate-engineered microbeads on the nuclear maturation and mitochondrial distribution patterns of prepubertal ovine oocytes.

Culture Condition	No. of Cultured COCs	No. of Analyzed COCs	Nuclear Chromatin Configurations Number (%)				P/S Mitochondrial Pattern (*)
			GV	MI to TI	MII	Abnormal	
2D	172	163	30 (18.4)	32 (19.6)	74 (45.4) ^a^	27 (16.6) ^a^	26/46 (57) ^a^
3D	181	175	30 (17.1)	24 (13.7)	106 (60.6) ^c^	15 (8.6) ^b^	58/83 (70) ^b^

Table legend: GV = germinal vesicle; M = metaphase; T = telophase. Chi-squared test: within each column, different superscripts indicate statistically significant differences: within each group, ^a,b^ = *p* < 0.05; ^a,c^ = *p* < 0.01. (*) Data refer to MII oocytes from 6–8 runs of IVM.

**Table 2 cells-10-01458-t002:** Effects of a 3D-IVM system in alginate-engineered microbeads on the developmental competence of prepubertal lamb oocytes.

Culture Condition	No. of Inseminated Oocytes	No. of Zygotes Evaluated after IVF	Embryo Developmental Stages Number (%)						Total Cleavage Number (%)
			2- to 3- Cell	4- to 7- Cell	8- to 15- Cell	16- to 31- Cell	Morula (>32 Cells)	Blastocyst (>64 Cells)	
**2D**	163	157	4 (2.5) ^a^	22 (14.0)	37 (23.6)	19 (12.1)	4 (2.5)	5 (3.2)	91 (57.9) ^a^
**3D**	175	172	25 (14.5) ^c^	34 (19.8)	30 (17.5)	16 (9.3)	4 (2.3)	9 (5.2)	118 (68.6) ^b^

Chi-squared test: within each column, different superscripts indicate statistically significant differences; ^a,b^: *p* < 0.05; ^a,c^: *p* < 0.001.

**Table 3 cells-10-01458-t003:** Effects of 3D millifluidic IVM culture in alginate-engineered microbeads on the maturation rate of prepubertal ovine oocytes.

Culture Condition	No. of Cultured COCs	No. of Analyzed COCs	Nuclear Chromatin Configurations Number (%)				P/S Mitochondrial Pattern (*)
			GV	MI to TI	MII	Abnormal	
3D	180	171	35 (20.5)	23 (13.5)	99 (57.9)	14 (8.2)	26/51 (51)
3D Millifluidic	177	170	50 (29.4)	15 (8.8)	91 (53.5)	14 (8.2)	32/47 (68.1)

Table legend: GV = germinal vesicle; M = metaphase; T = telophase. Chi-squared test: NS. (*) Data refer to MII oocytes from four runs of IVM.

**Table 4 cells-10-01458-t004:** Effects of granulosa cell and type I collagen addition on the maturation rate of prepubertal ovine oocytes cultured under 2D- or 3D-IVM methods.

IVM Method	GC Culture Condition	Collagen I	No. of Cultured COCs (*)	No. of Analyzed COCs	CC Apoptotic Index (%)	Nuclear Chromatin Configurations Number (%)					
						GV	MI to TI	MII	Activated	Abnormal	P/S Mitochondrial Pattern
2D	-	-	178 (7)	123	24 ^a^	19 (15) ^a^	10 (8)	61 (50) ^a^	3 (2)	30 (25) ^c^	16/53 (30) ^a^
	Monolayer on uncoated wells	-	182 (7)	139	37 ^b^	36 (26) ^b^	10 (7)	62 (45)	3 (2)	28 (20)	13/56 (24)
3D	-	-	166 (7)	148	24 ^a^	29 (20) ^x^	15 (10) ^x^	93 (63) ^b,c^	2 (1)	9 (6) ^c,d^	44/86 (51) ^b^
	Included in microbeads	-	136 (8)	126	37 ^b^	45 (36) ^y^	3 (2) ^y^	32 (25) ^d,x^	4 (3)	42 (33) ^x,d^	12/32 (38) ^a^
	-	In microbeads	59 (3)	59	43 ^b, e^	16 (27)	0 (0)	32 (54)	4 (7)	7 (12)	18/32 (56)
	Included in microbeads	In microbeads	46 (3)	45	29 ^f^	15 (33)	0 (0)	29 (64) ^y^	0 (0)	1 (2) ^y^	18/26 (69) ^b^
	Monolayer on CI-coated wells	As plate coating	70 (3)	69	U.D.	17 (25)	0 (0)	46 (66) ^y^	2 (3)	4 (6) ^y^	19/46 (41)

Table legend: COC = cumulus–oocyte complex; GC = granulosa cell; GV = germinal vesicle; M = metaphase; T = telophase; P/S = perinuclear/subplasmalemmal mitochondrial distribution pattern. (*) In brackets, numbers of replicates are indicated. U.D. = unavailable data. Chi-squared test: within each column, different superscripts indicate statistically significant differences: ^a,b^ = *p* < 0.05; ^e,f^ = *p* < 0.05; ^x,y^ = *p* < 0.01; ^c,d^ = *p* < 0.001.

## Data Availability

Not applicable.

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
