# Peer review of "Bioengineering Approaches to Improve In Vitro Performance of Prepubertal Lamb Oocytes"

_cells, 2021, doi:10.3390/cells10061458_

Round 1
Reviewer 1 Report
This paper is addressing the complexity of in vitro maturation of mammalian oocytes. The methods are well described and adds further to the benefits of 3-D culture systems.
However, both the introduction and the discussion could be shortened and made more direct to the conclusions
Author Response
We thank the reviewer for suggestions which gave us the opportunity to improve the quality of the manuscript.
The English language has been revised, as required.
Introduction and Discussion have been shortened and made more direct, as required.
A relevant reference has been added.
Results presentation has been improved, as requested.
Reviewer 2 Report
Please see Note Comments in attached PDF

Author Response
We thank the reviewer for suggestions which gave us the opportunity to improve the quality of the manuscript.
The English language of the manuscript has been revised, as required.
Results presentation has been improved, as requested
Comments on attached file (line number referred to the submitted manuscript):
Lines 33-38: this run-on sentence has been broken into multiple sentences, as requested.
Line 46: the word “amply” has been removed.
Lines 54-60: this sentence has been restructured to reduce combo of ; and ,.
Lines 302-309: we thank the reviewer for positive comments
Line 326: Table 1 and Table 2 have been re-configured to remove word hyphenation.
Lines 339-344 and 373-384: Figure 2 and Figure 4 legends have been corrected.
Lines 363-367 (Figure 3): A double check of the statistical analysis was performed, as requested, by using the Unpaired Student’s t test: significance of blastocyst apoptotic index was confirmed whereas those of diameter and number of nuclei were not confirmed. The related part of the text and Figure 3 have been modified accordingly. Since the diameter and number of nuclei data are higher in group “3D” we decided to leave the related parts in “figure 3” to show the increasing trends.
Line 414: The word “computed” has been added.
Linea 529: the word “proven” has been replaced with “demonstrated”.
Reviewer 3 Report
please see attached

Author Response
Overview and originality
In this paper, COCs were cultured in microbeads, with and without micro-fluidics (using a ‘Live Box reactor), with and without granulosa cells and were compared to more traditional 2D culture systems, also using collagen plating. The system was testing COCs from Juvenile lambs, and was following on from a proof of concept of the technique using adult lambs.
While the concept of a 3D system is interesting, the results indicating the positive effects of 3D culture were not convincing and at times unclear. Also, the manuscript would need editing for correct spelling and English grammar. Suggest authors try to narrow the scope and add more data- perhaps focussing on the bioreactor as the novelty factor. While alginate beads maybe useful for follicle growth, I do not believe there is great improvement just for IVM.
We thank the reviewer for these comments which gave us the opportunity to improve the quality of our manuscript.
The English language of the manuscript has been revised, as required.
The text has been modified to narrow the scope and to focus on main discussion points.
We performed another 3D-IVM replicate of bioreactor. Additional data were provided in Table 3.
Concerning utility of alginate beads in follicle versus oocyte in vitro culture, we agree with the reviewer about the importance of their use for follicle growth (reviewed by Mastrorocco et al., 2020). However, this is a long-lasting culture method in which oocytes have to reach their somatic growth. 3D-IVM could also be very important for several applications in human and animal ARTs. Indeed, COC microencapsulation may prevent its flattening on the bottom of the plate, typical of a traditional 2D culture system, which causes reduction of COC viability due to limited acquisition of nutrients as about 50% of the cell surface is exposed to medium. In 3D-IVM, the goal is to improve the competence of fully grown oocytes immediately available for IVF or ICSI. By this way, it is possible to obtain matured oocytes from the ovaries (also criopreserved) of endangered animal species or autochthonous animal breeds or human oncological patients, for their immediate use in ART programs.
I mention a few doubts on the scientific validity of the paper below:
My main confusion was what gas was being used. For the 3D culture, it states a ‘CO2 atmosphere’ _– _is this 5% CO2 in air?
Later for the microfluidics (beads in the bioreactor) there is discussion of trying to get O2 to a 5% level, and testing O2 levels within the bioreactor and the normal dish. There was a distribution of high to low levels (0.05mol/m3 to 0.005mol/m3) but the gas flowing into the system wasn’t specified clearly—was it 5%CO2, 5% O2, 90%N2 or was it 5% CO2 in air?
All experiments were performed in an incubator with 5%CO2 in air, with the exception of in vitro embryo culture which was performed under 5%CO2, 5%O2 and 90% N2. We clarified this aspect in Materials and Methods section.
For the computational study, we modeled hypoxic conditions for further applications. We clarified also this point, including the results of normoxic culture conditions.
Blastocyst numbers for the overall study were very low (5 for 2D and 9 for 3D), thus making meaningful comparisons of blastocyst expansion, apoptotic indices invalid. We all know that blastocysts vary in size, just from one drop to another. More numbers are required, also you could look at mitotic division as an indicator of health also.
Also M11 rates were not high for any group. Maturation and cleavage stage rates for the normal 2d system for prepubertals is much lower than previously published -even over 15 years ago. (eg 69% cleavage and 43% blast rates from prepubertal lamb ovaries- Theriogenology 64 (2005) 1320–1332), thus reporting improvements with the 3D system is not that valid.
We agree with the reviewer about the fact that some previous studies reported better cleavage and blastocyst formation rates with prepubertal lamb oocytes, such as Morton et al., (2005).
However, different experimental conditions, necessarily used in our laboratory, may have affected oocyte maturation and fertilization and embryo development rates, such as IVM medium supplementation with FCS instead of estrous sheep serum, different LH/FSH preparations and, very importantly, frozen ram sperm instead of fresh one.
Nevertheless, in each experiment we reported statistically significant differences in our examined conditions compared with controls.
Although estrous sheep serum and fresh ram sperm increase the embryo cleavage and blastocyst formation rate, they also increase the experimental variability. Instead, FCS and frozen sperm allowed to observe the effects of analyzed methods under standardized conditions.
The statistical analysis concerning blastocyst quality has been double checked by using the Unpaired Student’s t test: significance of blastocyst apoptotic index was confirmed whereas those of diameter and number of nuclei were not confirmed. The related part of the text and “Figure 3” have been modified accordingly. Since the diameter and number of nuclei data are higher in group “3D” we decided to leave the related parts in “figure 3” to show the increasing trends. Based on data of reduced apoptotic index and improved bioenergetic/oxidative status, we can conclude that 3D-IVM improved the quality of developing embryos.
Additional data have been provided after IVM in bioreactor experiment, as requested.
Further studies are in program to increase data on the effects of tested bioengineering approaches on embryo cultures.
The mitochondrial images are also unconvincing. Need to be much higher magnification to detect changes, especially where the mitochondria are wrt to the nucleus.
The quality of mitochondrial images has been improved, as requested.
As reported in methods, the mitochondrial distribution pattern is observed under confocal laser scanning microscopy (630x magnification) by direct analysis of 25 focal planes.
Therefore, the modifications are evaluated as a global assessment of observations all planes. The perinuclear distribution refers to the hemisphere in which the meiotic spindle is present. Pictures were always taken at the equatorial plane (which is the most representative). Quantification analysis of oocytes is performed on the equatorial plane whereas those of blastocysts are performed including all 25 planes in order to avoid variability due blastocoelic cavity morphology.
In figure 9, images at low brightness express results also reported in the graphs.